# Two-Dimensional Indium Selenide for Sulphur Vapour Sensing Applications

**DOI:** 10.3390/nano10071396

**Published:** 2020-07-18

**Authors:** Daniel Andres-Penares, Rodolfo Canet-Albiach, Jaume Noguera-Gomez, Juan P. Martínez-Pastor, Rafael Abargues, Juan F. Sánchez-Royo

**Affiliations:** 1ICMUV, Instituto de Ciencia de Materiales, Universidad de Valencia, P.O. Box 22085, 46071 Valencia, Spain; Daniel.Andres@uv.es (D.A.-P.); Rodolfo.Canet@uv.es (R.C.-A.); Jaume.Noguera@uv.es (J.N.-G.); Juan.Mtnez.Pastor@uv.es (J.P.M.-P.); rafael.abargues@uv.es (R.A.); 2MATINÉE: CSIC Associated Unit-(ICMM-ICMUV of the University of Valencia), Universidad de Valencia, P.O. Box 22085, 46071 Valencia, Spain

**Keywords:** two-dimensional semiconductors, III-VI semiconductors, InSe, photoluminescence, vapour sensing, chemical sensor

## Abstract

Surface-to-volume ratio in two-dimensional (2D) materials highlights among their characteristics as an inherent and intrinsic advantage taking into account their strong sensitivity to surface effects. For this reason, we have proposed in this work micromechanically exfoliated 2D nanosheets of InSe as an optical vapour sensor. As a proof of concept, we used 2-mercaptoethanol as the chemical analyte in vapour phase to monitor the change of the InSe photoluminescence (PL) before and after exposure to the analyte. For short vapour exposure times (at low analyte concentration), we found a PL enhancement of InSe nanosheets attributed to the surface localization of Se defects. For long vapour exposure times (or higher concentrations) a PL reduction is observed, probably due to the diffusion of molecules within the nanosheet. These results confirm the capability of 2D InSe as a photoluminescent sensor of vapours, because of its sensitivity to surface passivation or volume diffusion of molecules.

## 1. Introduction

Two-dimensional (2D) materials have been vastly studied in the last decade since graphene was first obtained [1] by micro-mechanical exfoliation from three-dimensional layered materials [2] or grown through methods like chemical vapour deposition (CVD) [3]. Among these new 2D materials, particular attention has been paid to transition metal dichalcogenides (TMDs) [4,5], hexagonal boron nitride (hBN) [6,7], and III-VI semiconductors [8,9,10]. Each one of these 2D materials has different properties from those of their bulk counterpart, due to quantum-confinement and surface effects [8,10,11], with promising opportunities in the fields of nano-optoelectronics [12,13,14], photovoltaics [15], optics [16], and energy storage applications [17,18].

Among 2D materials, indium selenide (InSe), a III-VI semiconductor with high electron mobility compared to other layered materials and potential applications in photovoltaics [19,20,21] acquires special relevance due to its highly tuneable bandgap from bulk to its monolayer [8,11]. This property, coupled with its high electron mobility [22], its recently demonstrated out-of-plane dipole luminescent orientation [23], and the relatively high reactivity of selenium, make 2D InSe a promising material for optoelectronic applications related to sensing. Generally speaking, the most valuable characteristic of a chemical sensor is its ability to provide a fast and selective response to a given analyte. The small size and high surface-to-volume ratio of 2D materials appears to be ideal for the design of fast sensors. In fact, some works have already reported 2D materials such as graphene oxide [24], TMDs [25], or phosphorene [26] for biological [27,28] and chemical sensing [26,29]. Among them, it is worth noting chemical sensing using optical instead of electrical properties of 2D materials as a transduction mechanism [30,31,32].

Focusing, then, on vapour sensing due to its more direct applicability, there are many interesting analytes for sensing. Among them, sensing of sulphur containing compounds are of particular interest for monitoring organic decomposition [33,34] or battery malfunction [35], which could be interesting for the food and energy storage industry. Similarly, sensing of nitro-containing compounds (-NO_2_) are of interest in the detection of explosives and analogues that has become a strategic priority in homeland security, land mine detection, and military issues [36].

In this work exfoliated InSe nanosheets have been demonstrated as a vapour sensor, which is validated by using 2-mercaptoethanol (MET) as an analyte, because thiols show a very strong affinity to metals [30,31,32]. After the characterization of the samples, we demonstrate chemosensing capabilities by measuring changes in the photoluminescence (PL) as a function of the exposure time to MET. We observe a fast sensor response time within tens of seconds. Both enhancement and quenching of PL is observed depending on the exposure time. A possible explanation of the underlying mechanisms behind these effects will be explored.

## 2. Materials and Methods

Bulk InSe monocrystals used to obtain the nanosheets were acquired through a perpendicular cleave of the (001) direction from an ingot grown by the Bridgman method [37] from a nonstoichiometric In_1.05_Se_0.95_ melt. Tin was added to act as a n-dopant in a content 0.01% previously to growth. MET was purchased from commercial suppliers and used as received (C_2_H_6_OS, Sigma Aldrich, St. Louis, MO, USA, 99%).

InSe nanosheets were obtained through mechanical exfoliation of 3D bulk InSe crystals. Once exfoliated, they were directly transferred from the scotch tape upon a 285 nm layer of silicon dioxide over silicon substrate, previously cleaned with acetone, ethanol, and isopropanol in an ultrasound bath.

To characterize the samples, two methods were used. Starting with the optical contrast analysis of the samples [7,38,39] to achieve a first thickness estimation on a Zeiss (Oberkochen, Germany) Axio Scope.A1 microscope with an Axiocam ERc 5s Camera. These measurements were corroborated using micro-photoluminescence (µ-PL) measurements, as the PL response of the material depends on its thickness, being a more reliable but much more time-consuming method than the first [11]. These measurements were taken in a Horiba (Kioto, Japan) Scientific Xplora setup, using a 532 nm CW excitation laser, not exceeding 70 µW of power in 7s acquisition time measurements in InSe nanosheets to prevent overheating. The optical excitation and collection spots are typically around 1 µm^2^. It is interesting to note here that we have maintained the same focal distance for PL spectra acquired in the 2D semiconductor before and after the vapour exposure in order to be able to perform a more direct quantitative comparison between intensities.

## 3. Results and Discussion

2D InSe nanosheets prepared by micromechanical exfoliation were tested as optical sensors, in order to be sensitive to local changes in the environment atmosphere when the target analyte molecules are chemisorbed on their surface. To validate the use of 2D InSe as a potential chemosensor, MET was selected as analyte. Thiol groups (–SH) of MET are very well-known chelating ligands. They exhibit a very strong affinity via their lone pair electrons of S towards metals such as indium atoms [40,41]. The 2D InSe sensor response was determined by monitoring changes in the PL response upon exposure to vapours of MET, as illustrated in Figure 1. The sensing protocol consisted of the exposure of 2D InSe nanosheets to the resulting analyte vapour from 40 mL of a 0.1 M MET aqueous solution, in a 100 mL closed vessel at room temperature. This concentration was chosen to follow more accurately the real-time kinetic sensor response in contrast to their direct exposure to vapours of pure MET (as can be observed in Appendix A for 99% purity MET), where a very strong response is observed after only 30 s. The MET concentration in vapour phase (see Table 1) can be estimated by Raoult’s Law (p_a_ = p*_a_ · x_a_, where pa is the partial pressure of the analyte in the solution, p*_a_ is the vapour pressure of the pure analyte, and x_a_ is the mole fraction of the analyte in the mixture), assuming that 0.1 M MET solution behaves as a thermodynamically ideal solution. Once prepared, all the sensing test follows the same protocol: 2D InSe samples (nanosheets deposited on the Si/SiO_2_ substrate) were attached to the inner side of the cap of a sealed vessel leaving the substrate faced-down and exposed directly for a given time. The exposure times to the analyte in vapour phase were systematically varied from 30 to 300 s. Given that the liquid-vapour equilibrium is broken each time that an exposure is carried out, so for every sample and measurement it was necessary to have an individual vessel with the same analyte in the required concentration, in order to assure the equilibrium vapour pressure, that needs hours to be reached. Furthermore, a constant concentration of the analyte in solution is assumed throughout the experiment because the analyte solution was saturated. The InSe nanosheets were characterized by µ-PL before and after exposure to the analyte by using the same PL setup and experimental conditions. Although InSe is stable in air [42], in order to prevent as much exposure to air as possible, the whole experimental process (vapour exposure and optical characterization) for each sample has been performed in a time span less than 4 h.

Figure 2 summarises the 2D InSe sensor response upon exposure to MET vapour. Figure 2b shows the µ-PL spectra acquired in the 2D InSe sample depicted in Figure 2a before and after 30 s exposure to MET 0.1 M, yielding a 1.5-fold PL enhancement in intensity. A similar measurement process was carried out in more than 60 different InSe nanosheets of different thicknesses (from 4 to 18 nm, as determined by the PL emission peak position measured in the exfoliated nanosheets [8]), collecting the change in the intensity of PL as the ratio I_t_/I_0_, where I_0_ and I_t_ is the PL maximum intensity before and after exposure at a given time. Figure 2c shows the real-time sensor response to vapours of MET.

Real-time sensor response (averaged for all measured InSe samples) in Figure 2c evidence two different trends, depending on the exposure time. At early stages of the exposure to MET, the PL intensity tends to increase, reaching a maximum PL intensity enhancement up to ~ 50% for 60 s of exposure. For longer exposure times, however, the ability of MET analyte to enhance the PL signal of InSe nanosheets tends to be reduced progressively, and the PL integrated intensity of the nanosheets after 300 s exposure is clearly smaller than its value before vapour exposure. Since bonding of MET to metals have a strong covalent contribution, i.e., thiols are very strong ligands, this response as a sensor is not reversible.

Our results clearly demonstrate that the sensing response of the 2D InSe to the MET vapour depends on the nanosheet thickness. In Figure 3 we have split all measurements (Figure 2c) into two groups: PL intensity ratio after/before MET exposure in 2D InSe nanosheets thicker (Figure 3a) and thinner than 6.5 nm (Figure 3b), respectively. 2D InSe nanosheets thicker than 6.5 nm exhibit a low response to vapour exposure at short exposure times (PL ratio = 1.13 in average) and a strong PL signal attenuation (PL ratio = 0.7–0.8 in average) for longer vapour exposure times. These results can be interpreted in the following way. At early stages of vapour exposure, the MET is expected to bond, preferably, to In atoms where there exists Se defects or vacancies at the topmost Se monolayer [43,44,45], favouring electron-hole interaction non-occurring in these surface defects passivated by MET molecules and, therefore, an enhancement in its PL emission is expected due to its consequently reduced non radiative carrier trapping. However, increasing the vapour exposure (this is, sufficiently long exposure times to MET or higher MET concentrations in solution, as observed in Appendix A with 99% purity MET), MET molecules would be forced to diffuse inside the material (as occurs for semiconductor doping processes). As a result, in equilibrium, MET will be fixed, most probably, within the nanosheets leading to a hampering carrier recombination and the subsequent decrease in PL emission. Analyte binding between layers would affect locally InSe nanosheets crystallographic structure creating defects, and therefore, reducing mobility and recombination processes [31]. These two effects can be exploited, first, as a sensor to check the presence of analytes by the modification of the nanosheet surface and, second, as a method to passivate of surface defects (and the subsequent minority carrier lifetimes) in future developments of electronic and optoelectronic devices.

These two different effects can be observed in thicker nanosheets (Figure 3a) in contrast with thinner ones (Figure 3b). In this case, for the time exposure studied and due to their higher surface-to-volume ratio compared with thicker ones, MET bonding in Se surface defects is expected to be more relevant, and therefore, a more maintained and higher PL ratio enhancement is observed. After that, a hampering due to MET diffusion within the nanosheets becomes relevant, reducing PL ratio measured in the nanosheets.

The defect-inhibition process and subsequent vapour diffusion into the InSe layers described above place the relevant question related to the possibility that vapour diffusion could take place in a layer-by-layer fashion. This possibility is particularly relevant for fine-tuning of quantum confinement effects [8,46,47,48,49]. Clearly, such a study would imply a careful and gradual vapour exposure process. However, preliminary results obtained already reveal a layer-by-layer vapour diffusion into 2D InSe nanosheets. Figure 4 shows the µ-PL response observed in three of the more than 60 near-2D InSe samples exposed and characterized in which quantum-confinement effects appear to be fine-tuned by means of vapour exposure. In these 2D InSe samples, we observed a blueshift of the PL emission after exposure to MET vapour. This can be attributed to an effective reduction of one layer and, therefore, in its PL peak position [8]. This fact can attributed to the reactivity of the surface to the molecules in air, with high concentrations or exposure times, the effective thickness of the sample could be reduced in one layer, considering the top layer completely bonded to the analyte molecules in air, obtaining a shift in the PL emission of the resulting nanosheet. This effect, instead of occurring in a controlled layer-by-layer alongside the nanosheet can be observed first in small micrometric regions that can be measured within a confocal micrometric collection spot. These results can be observed in Figure 4.

Results reported here allow us to propose 2D InSe nanosheets as nanometric vapour sensors for sulphur-containing compounds like MET, as a representative example. Besides, we also found promising sensing response of 2D InSe nanosheets to nitro-containing compound characteristics of explosives like 3-nitrotoluene (3-NT) with very fast response as shown in Appendix A. We believe that the present work paves the path to different studies using this 2D material as building blocks for the fabrication of fluorescent/photoluminescent sensing devices.

## 4. Conclusions

In this work, we have studied the ability of 2D InSe nanosheets to act as fluorescent sulphur sensors. A change in the photoluminescence of exfoliated InSe nanosheets has been observed by the exposure to 2-mercaptoethanol due to the reactivity of surface exfoliated Se free radicals in InSe. Our results reveal that, at lower concentrations and exposure times, the vapour preferably bonds to surface defects or Se vacancies improving the electronic surface quality and enhancing the luminescent response of the nanosheet. However, higher vapour exposure times or concentrations, tends to favour vapour diffusion into the InSe nanosheets, which hampers its luminescent response. Our preliminary results show that vapour diffusion into 2D InSe tends to occur in a layer-by-layer fashion that introduces significant changes as a function of the nanosheet thickness.

## Figures and Tables

**Figure 1 nanomaterials-10-01396-f001:**
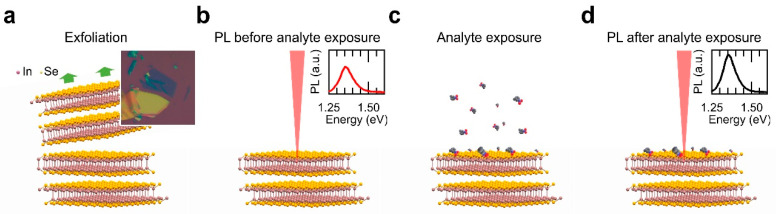
Schematic diagram of the exposure to vapour process followed in this work. After InSe cleavage (**a**), the nanosheet surface was characterized by photoluminescence (PL) (**b**), exposed to the analyte vapour in a controlled atmosphere (**c**), and again characterized by PL (**d**).

**Figure 2 nanomaterials-10-01396-f002:**
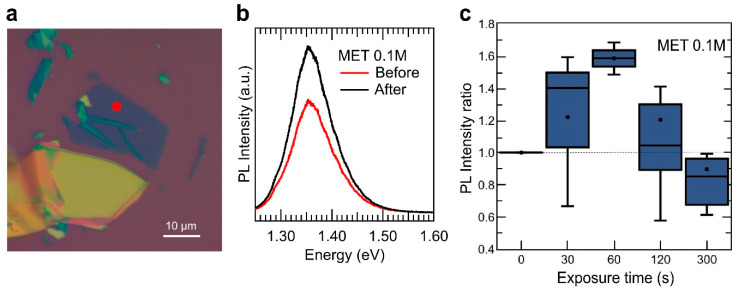
Vapour sensing with InSe. (**a**) optical image of a 2D InSe sample used in this experiment. Different optical contrast can be observed in the nanosheets for a fast thickness estimation (yellow for semi-bulk samples and deep blue to purple for thinner ones), marked with a red circle the position measured in an 8 nm nanosheet in (**b**) before and after vapour exposure. (**b**) micro-photoluminescence (µ-PL) spectra acquired in the exfoliated nanosheet shown in (**a**), as-exfoliated and after a 30 s exposure to 2-mercaptoethanol (MET) 0.1 M. (**c**) integrated PL intensity measured in several 2D InSe nanosheets as a function of exposure time to the vapour produced by MET 0.1 M in solution. The integrated PL signal has been normalized to the value measured in every sample before vapour exposure. For each exposure time, error bars, mean value (marked as an ×), median (horizontal line), and quartile calculation using inclusive median method (dark-blue rectangle) have been obtained using all the statistics collected for that specific exposure time.

**Figure 3 nanomaterials-10-01396-f003:**
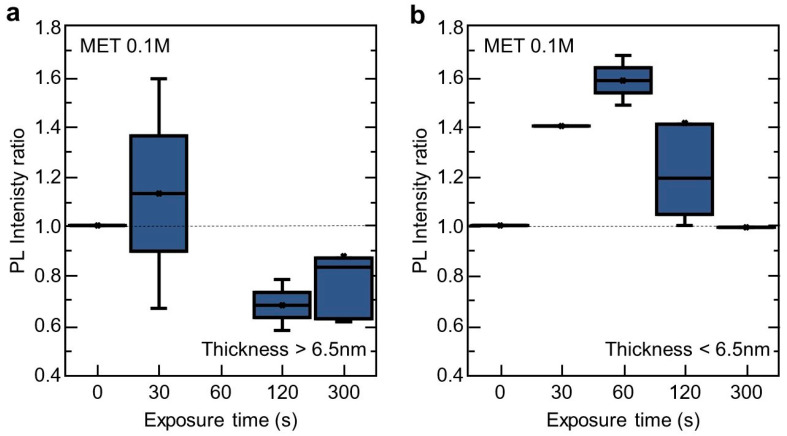
Vapour sensing ability of 2D InSe for different thickness. Ratio of integrated PL intensity after/before exposure to a MET vapour (from a solution 0.1 M) by InSe nanosheets thicker (**a**) and thinner (**b**) than 6.5 nm. For each exposure time, error bars, mean value (marked as an ×), median (horizontal line), and quartile calculation using inclusive median method (dark-blue rectangle) have been obtained using all the statistics collected for that specific exposure time.

**Figure 4 nanomaterials-10-01396-f004:**
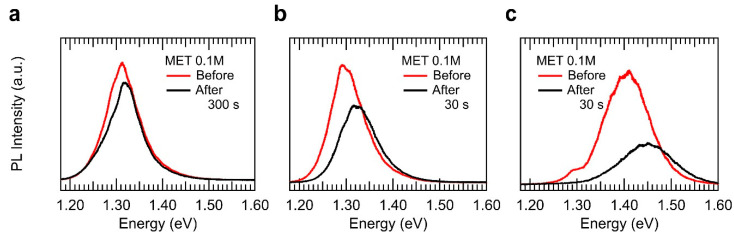
Layer-by-layer vapour sensing effects in 2D InSe. (**a**)–(**c**) three different examples of the enhancement of quantum confinement effects in 2D InSe nanosheets of thickness 16, 14, and 6 nm, respectively, observed as a consequence of the vapour exposure.

**Table 1 nanomaterials-10-01396-t001:** Concentration of vapour exposed to InSe nanosheets.

Analyte	[Analyte]_liq_ M	[Analyte]_vap_ M	[Analyte]_vap_ ppb
Water	-	-	-
2-mercaptoethanol ^1^	0.1	1.4 · 10^−7^	10.9
2-mercaptoethanol ^1^	16.8	8 · 10^−5^	5714

^1^ vapour pressure 1.5 mmHg at 20 °C.

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
