# Peer review of "Two-Dimensional Indium Selenide for Sulphur Vapour Sensing Applications"

_nanomaterials, 2020, doi:10.3390/nano10071396_

Round 1

Reviewer 1 Report

This paper investigates photoluminescence characteristics of 2D InSe nanosheets exposed to MET vapor and discusses their potential as a sulphur sensor. Interesting results were obtained, and the article is well organized. The paper will attract much attention from those who are working on 2D materials. Although this paper has great potential to be published in Nanomaterials, the following revisions are recommended:

  1. The reason for the different behaviors between Figs. 3(a) and (b) should be clarified. The explanation for Fig. 3(b) is not found, even though its behavior is quite different from that of Fig. 3(a).
  2. It would be helpful if the authors could elaborate on the reason that MET fixed within the nanosheet hampers carrier recombination (not non-radiative recombination). Can that be understood that the InSe layer is damaged and does not work as an InSe layer, as mentioned, “an effective reduction of one layer”?
  3. Are the peak energy shifts in Fig. 4 quantitatively consistent with those for the one layer reduction?
  4. It would be helpful if the authors could explain Fig. 2(a) in more detail. Is the deep blue layer an InSe layer? What is the yellow layer? Is the red point a laser spot? What is the thickness of InSe for Fig. 2(b)?
  5. PL decreases when MET is diffused and fixed within the nanosheet.
  6. Is the response of InSe to the exposure of MET reversible? Does the PL intensity recover after stopping exposure to MET?
  7. Is the sensitivity of 2D InSe to sulphur sufficiently high for practical use and higher than that of other conventional sulphur sensors?
  8. “2-mercaptoethanol (MET)” in line 82 on page 2 should be “MET.”

Author Response

Below can be found the Reviewer comment with our point-by-point response in bold letters.

This paper investigates photoluminescence characteristics of 2D InSe nanosheets exposed to MET vapor and discusses their potential as a sulphur sensor. Interesting results were obtained, and the article is well organized. The paper will attract much attention from those who are working on 2D materials. Although this paper has great potential to be published in Nanomaterials, the following revisions are recommended:

  1. The reason for the different behaviors between Figs. 3(a) and (b) should be clarified. The explanation for Fig. 3(b) is not found, even though its behavior is quite different from that of Fig. 3(a).

In order to clarify this explanation, the next paragraph has been included after line 163 (new line 175).

These two different effects can be observed in thicker nanosheets (Figure 3a) in contrast with thinner ones (Figure 3b). In this case, for the time exposure studied and due to their higher surface – to – volume ratio compared with thicker ones, MET bonding in Se surface defects is expected to be more relevant (and therefore, it is observed a more maintained and higher PL ratio enhancement). After that, a hampering due to MET diffusion within the nanosheets becomes relevant, reducing PL ratio measured in the nanosheets.

  1. It would be helpful if the authors could elaborate on the reason that MET fixed within the nanosheet hampers carrier recombination (not non-radiative recombination). Can that be understood that the InSe layer is damaged and does not work as an InSe layer, as mentioned, “an effective reduction of one layer”?

In order to clarify this explanation, the following line has been included after line 160 (new line 170).

Analyte binding between layers would affect locally InSe nanosheets crystallographic structure creating defects, and therefore, reducing mobility and recombination processes.[31]

[31] Rodríguez-Cantó, P.J.; Abargues, R.; Gordillo, H.; Suárez, I.; Chirvony, V.; Albert, S.; Martínez-Pastor, J. UV-patternable nanocomposite containing CdSe and PbS quantum dots as miniaturized luminescent chemo-sensors. RSC Adv. 2015, 5, 19874–19883, doi:10.1039/c4ra02812k.

  1. Are the peak energy shifts in Fig. 4 quantitatively consistent with those for the one layer reduction?

Results showed in Figure 4 represent the PL shift after vapour exposure in three different nanosheets with different thicknesses (16, 14 and 6 nm). In InSe nanosheets, one-layer reduction can be observed in both a blue shift and an intensity reduction in the PL [8] but due to the effect of the vapour in the PL intensity emission in our project, only the PL position can be used to determine its thickness. These shifts when the thickness is reduced in one layer are larger for thinner samples, where a distinguishable positions when one layer is reduced can be completely separated (as observed in Figure 4c, with a shift from 1.40 to 1.45 eV reproduces undoubtedly the transition from 6nm to 5nm with one-layer reduction). However, for thicker near-2D samples (for example, in Figure 4a and b, with thicknesses of 16 and 14nm, respectively), these shifts are narrower when one single layer is reduced, observing typical shifts within this thickness range of 0.015eV. This PL-thickness estimation goes correlated with an intensity decrease, which complement the analysis, that cannot be done in our case due to the additional vapour effect previously studied, so in these semi-bulk cases, a precise one-layer reduction cannot be assured as strictly as before with thinner samples. An effective thickness reduction is clearly observed due to this shift but can be associated in Figure 4a and b to one-layer to three-layer reduction. This discussion has not been included in the present work because further analysis and measurements have to been performed to precisely understand this observed behaviour prior to any conclusion explaining, therefore, the one-layer effective reduction as a first step that can be accumulated consecutively in thicker nanosheets.    

  1. Brotons-Gisbert, M.; Andres-Penares, D.; Suh, J.; Hidalgo, F.; Abargues, R.; Rodríguez-Cantó, P.J.; Segura, A.; Cros, A.; Tobias, G.; Canadell, E.; et al. Nanotexturing to Enhance Photoluminescent Response of Atomically Thin Indium Selenide with Highly Tunable Band Gap. Nano Lett. 2016, doi:10.1021/acs.nanolett.6b00689.

  1. It would be helpful if the authors could explain Fig. 2(a) in more detail. Is the deep blue layer an InSe layer? What is the yellow layer? Is the red point a laser spot? What is the thickness of InSe for Fig. 2(b)?

To clarify that figure, the caption in Figure 2a has been modified to

(a) Optical image of a 2D InSe sample used in this experiment. Different optical contrast can be observed in the nanosheets for a fast thickness estimation (yellow for semi-bulk samples and deep blue to purple for thinner ones), marked with a red circle the position measured in an 8 nm nanosheet in (b) before and after vapour exposure.

  1. PL decreases when MET is diffused and fixed within the nanosheet.

Precisely, that is one of the effects that appears specially in thicker nanosheets, but also in thinner ones at longer exposure time (reason why the PL ratio starts decreasing after 60s). This diffusion and embeddedness effect that hampers the PL after exposure compete with the first surface localisation of Se defects that enhances their PL at lower exposure time.

  1. Is the response of InSe to the exposure of MET reversible? Does the PL intensity recover after stopping exposure to MET?

To clarify this question, it has been included in line 144 the following sentence:

Since that bonding of MET to metals have a strong covalent contribution, i.e., thiols are very strong ligands, this response as a sensor is not reversible.

  1. Is the sensitivity of 2D InSe to sulphur sufficiently high for practical use and higher than that of other conventional sulphur sensors?

Further analysis and study are needed to obtain reliable sensitivity figures-of-merit in order to compare this material with conventional sulphur sensors (which have been present for decades, with constant improvement through the years until the commercial stage). In this manuscript we present the capabilities of InSe among the 2D semiconductors as a sensing platform for the first time in literature, and the advantages this material has for specific applications due to its intrinsic characteristics.

  1. “2-mercaptoethanol (MET)” in line 82 on page 2 should be “MET.”

Line 82 (new line 88) has been modify including the change proposed.

Reviewer 2 Report

This is an interesting paper describing photoluminescence property of InSe nano-sheet for Sulphur 2 vapour sensing applications.

I think that this paper requires major revisions to be accepted for publication in Nanomaterials.

Please refer to the comments below.

  1. In line 87 – 90, the authors described that “The sensing protocol consisted of the exposure of 2D InSe to the resulting analyte vapour from 40 mL of a 0.1 M MET aqueous solution, in a 100 mL closed vessel at room temperature. This concentration was chosen to follow more accurately the real-time kinetic sensor response in contrast to their direct exposure to vapours of pure MET (as can be observed in Suppl. Figure S1 for 99 % purity MET), where a very strong response is observed after only 30 s.” A series of PL property for 0.1 M MET aqueous solution was described in the manuscript, however, the experimental result of the PL property for 99 % purity MET is limited (one data shown only in Figure S1).  Readers do not know the sensor response upon the exposure of 99 % purity MET.  If authors want to discuss the comparison of PL property of two MET concentrations, please give a series of PL property (exposure time dependence on PL spectra / intensity) for 99 % purity MET.  If authors do not want to compare them, it is better to omit the data for 99% purity MET of Figure S1.
  2. The authors showed optical image of a 2D InSe sample in Figure 2 (a). Please explain what the color contrast means and where the boundary of the sample is in the image for readers’ understanding.
  3. The authors shows the PL spectra before and after the exposure of 0.1 MET in Figure 2. Please describe the thickness of the sample for these experiments.  Also, please add all the PL spectra for exposure time of 60, 120, 300 s, as well as the data before and after 30 s exposure.  The peak energy and FWHM (full width at half maximum) of the PL spectra provides the information for discussion of the mechanism, as are shown in Figure 4.
  4. Please add the data of the PL intensity ratio for exposure time of 60 s in the data of Figure 3(a), which is missing in the submitted manuscript and very important for investigation.
  5. The authors showed the comparison of PL spectra of the sample having different thickness before and after exposure of MET 0.1 M in Figure 4. PL spectra after 30 s (not 300 s) is required for comparison with the data of (b) and (c).
  6. The authors showed enhancement of quantum confinement effects in 2D InSe. From the viewpoint of the sensing applications, are there any optimum thickness providing the better sensitivity.  Please describe these points.
  7. The authors showed the vapour sensing for exposure to 3-NT 99 %. However the title of this paper is “Two-dimensional indium selenide for Sulphur vapour sensing applications”.  Since the data of 3-NT 99% is limited and out of the title, I strongly recommend that the data of 3-NT 99 % is prepared in future publication.

Author Response

Below can be found the Reviewers comments with a point-by-point response in bold letters.

This is an interesting paper describing photoluminescence property of InSe nano-sheet for Sulphur 2 vapour sensing applications.

I think that this paper requires major revisions to be accepted for publication in Nanomaterials.

Please refer to the comments below.

  1. In line 87 – 90, the authors described that “The sensing protocol consisted of the exposure of 2D InSe to the resulting analyte vapour from 40 mL of a 0.1 M MET aqueous solution, in a 100 mL closed vessel at room temperature. This concentration was chosen to follow more accurately the real-time kinetic sensor response in contrast to their direct exposure to vapours of pure MET (as can be observed in Suppl. Figure S1 for 99 % purity MET), where a very strong response is observed after only 30 s.” A series of PL property for 0.1 M MET aqueous solution was described in the manuscript, however, the experimental result of the PL property for 99 % purity MET is limited (one data shown only in Figure S1).  Readers do not know the sensor response upon the exposure of 99 % purity MET.  If authors want to discuss the comparison of PL property of two MET concentrations, please give a series of PL property (exposure time dependence on PL spectra / intensity) for 99 % purity MET.  If authors do not want to compare them, it is better to omit the data for 99% purity MET of Figure S1.

We believe that the exposure of the pure vapour data showed in Figure Suppl. 1 helps to understand how the vapour affects nanosheets in the case of higher exposition times (being equivalent at reduced exposure times with higher concentrations, as showed with the pure vapour). Using the exposure technique employed in this work, a more detailed time dependence for 99% purity MET is not possible due to experimental limitations. However, it is worth noticing that a further study in this direction could be utterly illuminating using a more precise exposure technique, for both time dependence at higher concentrations and a possible layer-by-layer effective bonding that has to be determined in the future.

  1. The authors showed optical image of a 2D InSe sample in Figure 2 (a). Please explain what the color contrast means and where the boundary of the sample is in the image for readers’ understanding.

To clarify that figure, the caption in Figure 2a has been modified to (a) Optical image of a 2D InSe sample used in this experiment. Different optical contrast can be observed in the nanosheets for a fast thickness estimation (yellow for semi-bulk samples and deep blue to purple for thinner ones), marked with a red circle the position measured in an 8 nm nanosheet in (b) before and after vapour exposure.

  1. The authors shows the PL spectra before and after the exposure of 0.1 MET in Figure 2. Please describe the thickness of the sample for these experiments.  Also, please add all the PL spectra for exposure time of 60, 120, 300 s, as well as the data before and after 30 s exposure.  The peak energy and FWHM (full width at half maximum) of the PL spectra provides the information for discussion of the mechanism, as are shown in Figure 4.

In Figure 2b a representative PL spectra before and after its exposure to the vapour can be seen as an example of the statistic that has been done using several nanosheets for different exposure times.  The experimental process has been the following: after a PL characterisation before the exposure, the nanosheet is introduced faced-down in a previously prepared 0.1M MET solution, leaving the nanosheet in a closed saturated vapour environment for the desired time. After that time, the PL of the nanosheet is measured again to obtain the PL after / before ratio. Therefore, one nanosheet is only used for one exposure time, not exposed through different exposure times. This procedure has been done to several nanosheets in the same conditions (same exposure time and vapour concentration) to have the statistic showed in the Figures 2c and 3. That is the reason why we have not compared different spectra of different nanosheets (which intensities can vary), but the ratios within the same nanosheet, using its PL intensity after and before exposure. The ratios (enhancement or hampering) can be compare in a time scale, unlike the direct PL intensity between nanosheets.

  1. Please add the data of the PL intensity ratio for exposure time of 60 s in the data of Figure 3(a), which is missing in the submitted manuscript and very important for investigation.

Due to the fact that each nanosheet has been exposed only to one exposure time, after separating all our data between thinner and thicker nanosheets, in that specific case no data can be showed. Besides, it is worth noticing that nanosheets that experienced a blueshift after their exposure also were removed (comparing an intensity change where two parameters can affect, the vapour bonding and a possible layer-to-layer effective thickness reduction is not possible at the moment, these cases have been highlighted in Figure 4). However, due to the fact that this time dependence comes after a more general one (Figure 2c, with all nanosheets included) and the trend is similar (it starts with a slight enhancement, to finish with a hampering), that point can be expected to be close to 1 as a turning point between these two effects.

  1. The authors showed the comparison of PL spectra of the sample having different thickness before and after exposure of MET 0.1 M in Figure 4. PL spectra after 30 s (not 300 s) is required for comparison with the data of (b) and (c).

As explained before, each nanosheet is only used for one exposure time, not exposed through different exposure times. That is the reason why for each panel, different times can be observed (being 300s, 30s and 30s, respectively for Figure 4a, 4b and 4c). In that cases, exposing these nanosheets to the specified time, the blueshift obtained is what can be seen in Figure 4.

  1. The authors showed enhancement of quantum confinement effects in 2D InSe. From the viewpoint of the sensing applications, are there any optimum thickness providing the better sensitivity.  Please describe these points.

This is a decision we believe has to be discussed for any specific application. One of the advantages of 2D InSe nanosheets that we tried to highlight was the usability of any thickness due to its band gap tunability. Therefore, for example, the optimum thickness could be determined for the PL sensor that has to be used for a specific energy or wavelength. If sensitivity is what is sought, in Figure 3 can be observed that thinner nanosheets present a more visible effect due to their higher surface – to – volume ratio, so in this case an InSe ML is expected to be the optimum, but limited to a 2.1 eV PL peak detection in this case, the PL of a InSe ML. Further studies can be done changing the exposure method to allow a real-time analysis, studying each thickness separately to determine each case possible.

  1. The authors showed the vapour sensing for exposure to 3-NT 99 %. However the title of this paper is “Two-dimensional indium selenide for Sulphur vapour sensing applications”.  Since the data of 3-NT 99% is limited and out of the title, I strongly recommend that the data of 3-NT 99 % is prepared in future publication.

Precisely, due to the fact that the whole analysis has been done using MET (a Sulphur compound), that is what we thought should be described in the manuscript title, just for Sulphur vapour sensing application. However, it is worth noticing that InSe nanosheets are promising not only for Sulphur, but Nitrogen compounds, more as a perspective for the scientific community, so other analysis can be developed in this case, and that is the reason why the data is showed only as a Suppl. Figure and only mentioned in the conclusions of our manuscript, aiming future works in this direction.

Reviewer 3 Report

In this manuscript, the authors have demonstrated Sulphur vapour sensing capabilities of InSe nanosheets by measuring changes in the photoluminescence (PL) as a function of the exposure time to 2-mercaptoethanol (MET). The research work carried out looks very interesting and promising vapour sensing applications.

Comments/Suggestions:

  1. Most of the PL measurement shown in the manuscript are before and after vapour exposure. Please show the cyclic response of the PL measurements when the same concentration of the MET is exposed to InSe nanosheets in a ON-OFF time. Please comment on the recovery time of the InSe nanosheets.
  2. The concentration of the vapour is varied based on the time exposure (30 to 300 s). Is it a right method? Please mention the tool used to know the exact concentration of vapour exposed on the surface of the InSe nanosheets.
  3. What will be the limit of detection (LOD) of our method?
  4. Please mention the right thickness of InSe nanosheets used in Figure 3, instead of mentioning Thickness > 6.5 nm and Thickness < 6.5 nm.
  5. In Figure 4a, I hope it is 30 s instead of 300 s. Please mention thickness in each graph.
  6. Please comment on the long-term repeatability characteristics of InSe nanosheets for vapous sensing. What will be the error bar of the PL measurement after one month with the same experimental procedure?

Author Response

Below can be found the Reviewers comments with a point-by-point response in bold letters.

In this manuscript, the authors have demonstrated Sulphur vapour sensing capabilities of InSe nanosheets by measuring changes in the photoluminescence (PL) as a function of the exposure time to 2-mercaptoethanol (MET). The research work carried out looks very interesting and promising vapour sensing applications.

Comments/Suggestions:

  1. Most of the PL measurement shown in the manuscript are before and after vapour exposure. Please show the cyclic response of the PL measurements when the same concentration of the MET is exposed to InSe nanosheets in a ON-OFF time. Please comment on the recovery time of the InSe nanosheets.

To clarify this question, it has been included in line 144 the following sentence: Since that bonding of MET to metals have a strong covalent contribution, i.e., thiols are very strong ligands, this response as a sensor is not reversible.

  1. The concentration of the vapour is varied based on the time exposure (30 to 300 s). Is it a right method? Please mention the tool used to know the exact concentration of vapour exposed on the surface of the InSe nanosheets.

In our measurements, the concentration of the vapour remains constant during the whole exposure time. The experimental process has been the following: after a PL characterisation before the exposure, the nanosheet is introduced faced-down in a previously prepared 0.1M MET solution, leaving the nanosheet in a closed saturated vapour environment for the desired time. After that time, the PL of the nanosheet is measured again to obtain the PL after / before ratio. Therefore, one nanosheet is only used for one exposure time, not exposed through different exposure times. This procedure has been done to several nanosheets in the same conditions (same exposure time and vapour concentration) to have the statistic showed in the Figures 2c and 3. That is the reason why we  can estimate the MET concentration in vapour phase (see Table 1) by Raoult’s Law due to the fact that the saturated concentration is not modify through the whole exposure time in any case.

  1. What will be the limit of detection (LOD) of our method?

We did not carry out the calibration curve which is necessary to determine the LOD. In fact LOD depends not only on the thickness of the 2D InSe but also on the lateral size of the sheets. Because we cannot produce 2D InSe with the same thickness and lateral size, measurement of the LOD would be very inaccurate.

  1. Please mention the right thickness of InSe nanosheets used in Figure 3, instead of mentioning Thickness > 6.5 nm and Thickness < 6.5 nm.

As explained before, for every point in that Figure several nanosheets have been analysed measured in the same conditions (exposure time and vapour concentration) to have the statistic to produce that Figures. Therefore, there is not one specific thickness for that Figures. The distinction there represents thinner and thicker nanosheets, where the surface – to – volume ratio changes drastically, producing different intensity in the effects described before. In concrete, for Figure 3, where nanosheets exposed to the diluted MET are showed separated due to their thicknesses, 10 nanosheets are thinner than 6.5 nm (PL emissions from 1.36eV to 1.48eV, corresponding to thicknesses from 6.5nm to 5nm) and 11 nanosheets are thicker than 6.5nm (with PL emissions from 1.28eV to 1.34eV, corresponding to thicknesses from 18nm to 7 - 8nm)

  1. In Figure 4a, I hope it is 30 s instead of 300 s. Please mention thickness in each graph.

For every sample it has been written the exposure time applied to that nanosheet, and it is correct that nanosheet showed in Figure 4a was exposed 30s meanwhile nanosheets showed in Figure 4b and 4c were exposed 300s. The thickness of each nanosheet can be seen in the caption “Three different examples of the enhancement of quantum confinement effects in 2D InSe nanosheets of thickness 16, 14 and 6 nm, respectively, observed as a consequence of the vapour exposure.”. For each nanosheet is only used for one exposure time, not exposed through different exposure times. That is the reason why for each panel, different times can be observed (being 300s, 30s and 30s, respectively for Figure 4a, 4b and 4c). In that cases, exposing these nanosheets to the specified time, the blueshift obtained is what can be seen in Figure 4.

  1. Please comment on the long-term repeatability characteristics of InSe nanosheets for vapous sensing. What will be the error bar of the PL measurement after one month with the same experimental procedure?

As reported in [38] and according to our own experience using this material, InSe exfoliated nanosheets are stable in air compared, for example, with GaSe nanosheets, and therefore its PL intensity is maintained stable several days after exfoliation. In our experiment, the whole experimental process (vapour exposure and optical characterization) has been developed in less than 4h, so any appreciable deterioration can be discarded. In our experience with InSe nanosheets, PL intensity can be maintained several weeks and months after exfoliation, so similar behavior can be expected if the nanosheets, instead of being freshly exfoliated, where exfoliated and used a month later.

  1. Del Pozo-Zamudio, O.; Schwarz, S.; Klein, J.; Schofield, R.C.; Chekhovich, E. a.; Ceylan, O.; Margapoti, E.; Dmitriev, a. I.; Lashkarev, G. V.; Borisenko, D.N.; et al. Photoluminescence and Raman investigation of stability of InSe and GaSe thin films. arXiv 2015, 1–6.

Reviewer 4 Report

This manuscript studied InSe as an optical vapor sensor, by monitoring PL signal intensity.

The trend of PL intensity behavior as exposure time of MET vapor is statistically studied and conclusive, and authors suggest a possible sensing mechanism, which is likely but needs further verification. 

However, the possiblity of "fine-tuning" of quantum confinment effects is interesting and supported by the blue shift in Fig. 4. 

This manuscript can be accepted after addressing the following practical questions as a vapor sensor. 

1) What is the motivation of detecting MET? In what purpose is the detection of MET useful for? 

2) What is the detection limit of MET? Is it sensitive enough for the purpose of the detection? Can the concentration of analytes be expressed in units of ppm or ppb instead of 40 mL of 0.1M solution, or vapor pressure?

3) During the vapor exposure, what gas is used as the background? Is it synthetic air, vacuum, or some innert gas?

4) Line 103-104: InSe is generally not stable in air. How stable is the device in air and how long can it be used?

5) In Fig. 3: How many devices are from less/thicker than 6.5 nm? If thicker than 6.5 nm, how thick is the thickest sample?

Author Response

Below can be found the Reviewers comments with a point-by-point response in bold letters.

This manuscript studied InSe as an optical vapor sensor, by monitoring PL signal intensity.

The trend of PL intensity behavior as exposure time of MET vapor is statistically studied and conclusive, and authors suggest a possible sensing mechanism, which is likely but needs further verification. 

However, the possiblity of "fine-tuning" of quantum confinment effects is interesting and supported by the blue shift in Fig. 4. 

This manuscript can be accepted after addressing the following practical questions as a vapor sensor. 

1) What is the motivation of detecting MET? In what purpose is the detection of MET useful for? 

MET is a thiol that represents Sulphur-containing compound. The interest in MET is based on the fact that they show a very strong affinity to metals such as In atoms, as cited in the manuscrip (line 82)

“Thiol groups (-SH) of MET are very well-known chelating ligands. They exhibit a very strong affinity via their lone pair electrons of S towards metals such as indium atoms”

Therefore, MET is an ideal molecule to carry out the proof of concept of InSe as a sensor.

On the other hand, 3-NT represents a Nitro-containing compound. In order to explain the motivation and applicability of these compounds, after line 48 the following paragraph has been included.

“Focusing then on vapour sensing due to its more direct applicability, there are many interesting analytes for sensing. Among them, sensing of sulphur containing compound are of particular interest for monitoring organic decomposition32,33 or battery malfunction34, which could be interesting for the food and energy storage industry. Similarly, sensing of nitro-containing compounds (-NO2) are of interest in the detection of explosives and analogues that has become a strategic priority in homeland security, land mine detection and military issues 35.

(32)           Taylor, W. F.; Wallace, T. J. Kinetics of Deposit Formation from Hydrocarbons: Effect of Trace Sulfur Compounds. Ind. Eng. Chem. Prod. Res. Dev. 1968, 7 (3), 198–202. https://doi.org/10.1021/i360027a009.

(33)           Haines, W. E.; Cook, G. L.; Ball, J. S. Gaseous Decomposition Products Formed by the Action of Light on Organic Sulfur Compounds. J. Am. Chem. Soc. 1956, 78 (20), 5213–5215. https://doi.org/10.1021/ja01601a022.

(34)           Wang, Q.; Zheng, J.; Walter, E.; Pan, H.; Lv, D.; Zuo, P.; Chen, H.; Deng, Z. D.; Liaw, B. Y.; Yu, X.; et al. Direct Observation of Sulfur Radicals as Reaction Media in Lithium Sulfur Batteries. J. Electrochem. Soc. 2015, 162 (3), A474–A478. https://doi.org/10.1149/2.0851503jes.

(35)           Miller, J. B.; Barrall, G. A. Explosives Detection with Nuclear Quadrupole Resonance: An Emerging Technology Will Help to Uncover Land Mines and Terrorist Bombs. Am. Sci. 2005, 93 (1), 50–57. https://doi.org/10.1511/2005.1.50

2) What is the detection limit of MET? Is it sensitive enough for the purpose of the detection? Can the concentration of analytes be expressed in units of ppm or ppb instead of 40 mL of 0.1M solution, or vapor pressure?

Further analysis and study are needed to obtain reliable sensitivity figures-of-merit and detection limit in order to compare this material with commercial sulphur sensors (which have been present for decades, with constant improvement through the years until the commercial stage). In this manuscript we present the capabilities of InSe among the 2D semiconductors as a sensing platform for the first time in literature, and the advantages this material has for specific applications due to its intrinsic characteristics.

Concerning the last question, in Table 1 a relation between M and vapour pressure can be seen. It has been added an additional column with ppb to clarify this issue (being 10,9 ppb (µg/L) for [Analyte]vap = 1.4 · 10-7 M and 5714 ppb for [Analyte]vap = 8 · 10-5 M.

3) During the vapor exposure, what gas is used as the background? Is it synthetic air, vacuum, or some innert gas?

We used atmospheric air during the vapour exposure to represent an average usage of the material.

4) Line 103-104: InSe is generally not stable in air. How stable is the device in air and how long can it be used?

As reported in [38] and according to our own experience using this material, InSe exfoliated nanosheets are stable in air compared, for example, with GaSe nanosheets, and therefore its PL intensity is maintained stable several days after exfoliation. In our experiment, the whole experimental process (vapour exposure and optical characterization) has been developed in less than 4h, so any appreciable deterioration can be discarded.

  1. Del Pozo-Zamudio, O.; Schwarz, S.; Klein, J.; Schofield, R.C.; Chekhovich, E. a.; Ceylan, O.; Margapoti, E.; Dmitriev, a. I.; Lashkarev, G. V.; Borisenko, D.N.; et al. Photoluminescence and Raman investigation of stability of InSe and GaSe thin films. arXiv 2015, 1–6.

5) In Fig. 3: How many devices are from less/thicker than 6.5 nm? If thicker than 6.5 nm, how thick is the thickest sample?

For this work, only InSe nanosheets with appreciable quantum confinement effects (i.e., a PL peak position blueshifted more than 0.03eV from the 1.23eV expected in bulk InSe) have been used for our analysis. The thickest nanosheets used in this project have been three nanosheets that showed a PL peak emission centered at 1.28 eV, corresponding to a thickness around 18nm. Concretely, for Figure 3, where nanosheets exposed to the diluted MET are showed separated due to their thicknesses, 10 nanosheets are thinner than 6.5 nm (PL emissions from 1.36eV to 1.48eV, corresponding to thicknesses from 6.5nm to 5nm) and 11 nanosheets are thicker than 6.5nm (with PL emissions from 1.28eV to 1.34eV, corresponding to thicknesses from 18nm to 7 - 8nm)

Round 2

Reviewer 2 Report

Based on the authors’ reply, I understand that this work is at early stage, and some future studies are required for optimizing the sensor capability.  At this stage, I think that the revised manuscript should be accepted for publication.